# Low Thermal Expansion Machine Frame Designs Using Lattice Structures

**Poom Juasiripukdee** [1,*] **, Ian Maskery** [2] **, Ian Ashcroft** [2] **and Richard Leach** [1]

1   Manufacturing Metrology Team, Faculty of Engineering, University of Nottingham,
    Nottingham NG8 1BB, UK; richard.leach@nottingham.ac.uk
2   Centre for Additive Manufacturing, Faculty of Engineering, University of Nottingham,
    Nottingham NG8 1BB, UK; ian.maskery@nottingham.ac.uk (I.M.); ian.ashcroft@nottingham.ac.uk (I.A.)
*   Correspondence: poom.juasiripukdee@nottingham.ac.uk; Tel.: +44-759-330-8925

**Abstract:** In this work, we investigated tessellating cellular (or lattice) structures for use in a low thermal expansion machine frame. We proposed a concept for a lattice structure with tailorable effective coefficient of thermal expansion (CTE). The design is an assembly of two parts: a lattice outer part and a cylindrical inner part, which are made of homogenous materials with different positive CTEs. Several lattice design variations were investigated and their thermal and mechanical performance analysed using a finite element method. Our numerical models showed that a lattice design using Nylon 12 and ultra-high molecular weight polyethylene could yield an effective in-plane CTE of $1 \times 10^{-9}$ K$^{-1}$ (cf. $109 \times 10^{-6}$ K$^{-1}$ for solid Nylon 12). This paper showed that the combination of design optimisation and additive manufacturing can be used to achieve low CTE structures and, therefore, low thermal expansion machine frames of a few tens of centimetres in height.

**Keywords:** coefficient of thermal expansion; lattice structure; metrology frame; finite element analysis; additive manufacturing





## 1. Introduction

Temperature fluctuations, even in controlled laboratory environments, can cause alignment drift in high-precision measuring machines, as their support frames undergo thermal expansion and contraction [1]. This can lead to significant measurement uncertainty and, therefore, represent a significant issue in precision engineering [2]. It is desirable to use structures with low, or ideally zero, coefficient of thermal expansion (CTE) in such systems, but low-CTE structures are difficult to design and fabricate [1]. The geometrical complexity enabled by additive manufacturing (AM) provides new opportunities [3–7] for low-CTE structures for precision measuring machines but, to the authors' knowledge, there is no established design route to achieve this.

The design of structures with low or tailorable CTE is a growing research topic, with applications identified in aerospace, optical measurement systems, and precision instruments [8–10]. For example, Sigmund et al. [11] proposed three-phase composites (Invar, nickel and voids) using topology optimisation to design structures with positive, zero, and negative CTE. Akihiro et al. [12] and Takezawa et al. [13] applied the same approach to the internal geometry of porous composites, resulting in planar negative CTE structures. Takezawa et al. [13] concluded that topology optimisation is the most effective structural design method for minimising thermally induced stress and maximising structural stiffness. Their structures had CTEs in the range of 300% to 1000% of those of the constituent materials, thus demonstrating that advanced computer design techniques can be used to create tailorable CTE structures [13].

Lakes, following another novel design route, based their tailorable-CTE structure on a repeating hexagonal lattice with bi-material ribs [14]. The ribs were made of bonded layers, resulting in bending under increasing temperature. From this concept, Lakes

also proposed a tetrakaidecahedral foam cell, resulting in large positive, zero, and large negative CTE structures. Wu et al. [15] similarly proposed two-dimensional (2D) and three-dimensional (3D) lattices with bi-material elements. Their 2D lattices had aluminium and copper as the constituent materials (with CTE = $2.3 \times 10^{-5}$ K$^{-1}$ and $1.85 \times 10^{-5}$ K$^{-1}$, respectively), resulting in an effective CTE of $-68.1 \times 10^{-6}$ K$^{-1}$ for the lattice structure.

An innovation in the design of low-CTE structures came from Jefferson et al. [16], who proposed a lattice with two dissimilar materials and an assembly step, in which the voids of a continuous honeycomb structure were filled with 'inserts' of a higher-CTE material; this formed a hybrid structure with a predicted CTE of close to zero. Bi-material solutions for low and tailorable CTE structures were examined extensively, with the results summarised in Table 1. Based on Table 1, previous bi-material lattice designs have been fabricated in a wide range of metals and polymers, and have provided effective CTE in the range of $-4 \times 10^{-4}$ K$^{-1}$ to $1 \times 10^{-3}$ K$^{-1}$.

The production of low-CTE bi-material lattice structures requires advanced manufacturing techniques or several sequential manufacturing steps. Such structures have been fabricated previously using multi-material AM processes [12,13,15,17–19], conventional manufacturing techniques followed by joining [15,17,20–22], and combinations of both techniques [8,23]. From these publications, it is clear that the designer of a low-CTE bi-material lattice needs to consider both the complexity and manufacturability of the lattice, as well as the challenge of assembly. Our proposed approach is to simplify as far as possible the design requirements for a low-CTE lattice by utilising high-CTE inserts of a simple cylindrical geometry, which are made from a commonly available polymer.

In this paper, we proposed a new and practicable design method for structures which provide low planar CTE, with an emphasis on meeting the performance requirements of a precision instrument support frame. We described the low planar CTE bi-material structural design concept, based on a repeating AM unit cell and high-CTE inserts. Our numerical modelling showed that planar tailored thermal expansion can be achieved by controlling the internal AM lattice geometry and through the suitable selection of the fabrication materials.

**Table 1.** Summary of current tailored thermal expansion lattice structures.

| Paper Reference | Structure Type | Material(s) | Effective CTE ($10^{-6}$ K$^{-1}$) |
|---|---|---|---|
| [8] | Lightweight cellular metal composites | Aluminium and Invar | −14 to 17.1 |
| [11] | Composites with extremal CTEs using topology optimisation | Invar and nickel | −4.97 to 35.0 |
| [12] | A porous material with planar negative CTE | VeroWhitePlus and TangoBlack Plus | −434 to 396 |
| [13] | Porous composites with tunable CTE | VeroWhitePlus and TangoBlack Plus | −300 to 1000 |
| [14] | A repeating hexagonal lattice with bi-material ribs | Two materials with CTE difference of $10^{-5}$ K$^{-1}$ | Large positive, zero, and large negative |
| [24] | A honeycomb lattice with bi-material ribs | Invar and steel | Zero |
| [15] | 2D and 3D lattices with bi-material elements | Aluminium and copper | −68.1 |
| [25] | Planar chiral lattices and cylindrical shells | Stainless steel 431 or Al7075, and Invar | −65.77 to 91.64 |
| [16] | A continuous honeycomb structure with inserts | Two different CTE materials | Near-zero |
| [17] | An Octet bi-materials | Al6061 and Ti–6Al–4V | 0.17 |

**Table 1.** *Cont.*

| Paper Reference | Structure Type | Material(s) | Effective CTE ($10^{-6}$ K$^{-1}$) |
|---|---|---|---|
| [26] | 2D metamaterials using bi-material re-entrant planar lattice structures | Stainless steel 431 or Al7075, and Invar | −3 to 2.5 |
| [27] | 3D metamaterials using bi-material re-entrant planar lattice structures | Stainless steel 431 or Al7075, and Invar | −8.69 to −5.22 |
| [28] | Micro-lattice composite structure | Two different CTE materials | Negative or zero |
| [20] | Stretch-dominated planar lattices with the low CTE with high stiffness | Al7075 and Ti–6Al–4V | Zero with high stiffness |
| [23] | Stretch-dominated planar lattices in the micro-scale (thin film) | Aluminium and titanium | −0.6 |
| [18] | 1D to 3D multi-stable architected materials with zero Poisson's ratio and controllable CTE | Polyamide 12 and glass beads reinforced polyamide 12 | Large positive, zero, and large negative |
| [29] | Lattice cylindrical shells with tailorable axial and radial CTE | Stainless steel 431 or Al7075, and Invar | −64.6 to 88.0 |

## 2. Methodology

### 2.1. Motivation and Lattice Design Concept

The motivating application for this investigation is a machine frame to support precision measurement equipment (e.g., cameras and optical projection systems). Such a frame would ultimately be used in a temperature-controlled metrology laboratory, with the temperature range around 19–23 °C. Figure 1 shows an example of the use of the (40 × 20 × 30) cm lattice with the precision measurement equipment. The thermal expansion of the frame over this temperature range should not be greater than the resolution of the precision measurement equipment [30], so it does not dominate measurement uncertainty for the instrument.

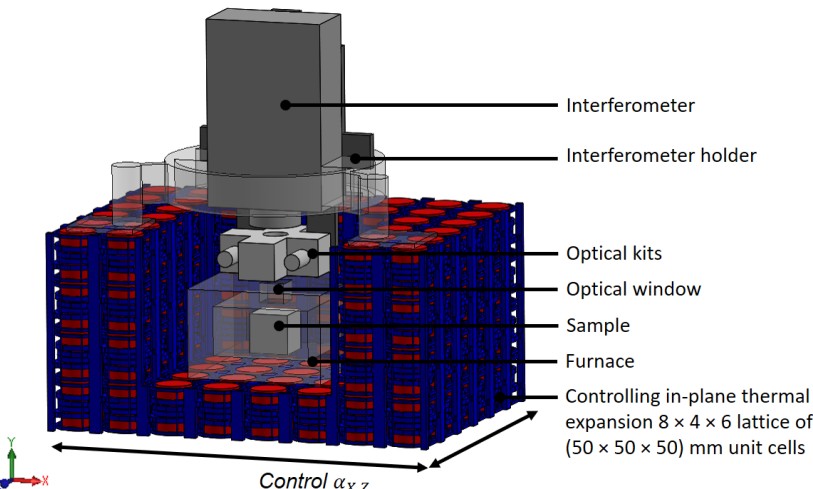

**Figure 1.** Illustration of precision measurement equipment with the 8 × 4 × 6 lattice; an optical vertical dilatometer.

To address the need of low planar CTE frames, lattice designs were developed and examined for their ability to provide a minimal, or zero, planar thermal expansion. A single-material lattice will generally possess similar CTE to a solid structure composed of the same material [31]. Thus, we focused on multi-material design solutions to obtain tailorable

thermal expansion. The principal challenge was how the internal lattice geometry could be created to minimise the structure's CTE using two different positive CTE materials with void spaces.

Initial concept lattice designs were modelled using computer-aided design (CAD) before being meshed and analysed using a finite element (FE) method. Mesh convergence analyses were carried out to ensure that the mesh density was sufficient to accurately predict the performance of the lattices. Using modelling and numerical analysis in this way provides a relatively inexpensive tool for the examination (and elimination) of concept designs in comparison with manufacture and physical testing. Finally, the most promising candidate design was examined in detail using computational methods prior to its manufacture and assembly being demonstrated.

The low-CTE lattice design was created as a unit cell which was then tessellated into a lattice structure (Figure 2). The unit cell was composed of a lattice outer part (marked 'Low-CTE' in Figure 2) and a cylindrical inner part (marked 'High-CTE'). The lattice outer part contained the CTE-minimising mechanism, and was intended to be made using an AM process, particularly polymer laser powder bed fusion (LPBF). The cylindrical inner part was selected from a range of conventional industrial materials and was, therefore, not restricted by the limited materials selection associated with LPBF. Polyamide 12 (Nylon 12) was chosen as a relatively low-CTE material for the lattice outer part because it processes well by LPBF to enable complex parts to be manufactured with a good combination of geometrical accuracy and mechanical performance [32]. Ultra-high-molecular-weight polyethylene (UHMWPE) was chosen as a relatively high-CTE material for the cylindrical inner part because it provides a high value of CTE, good impact resistance, very low coefficient of friction, and self-lubricating performance [33], which is advantageous in the assembly and performance of the proposed two-material structure. The two parts were manufactured separately then hand assembled by inserting the cylindrical inner part into the centre of the lattice outer part.

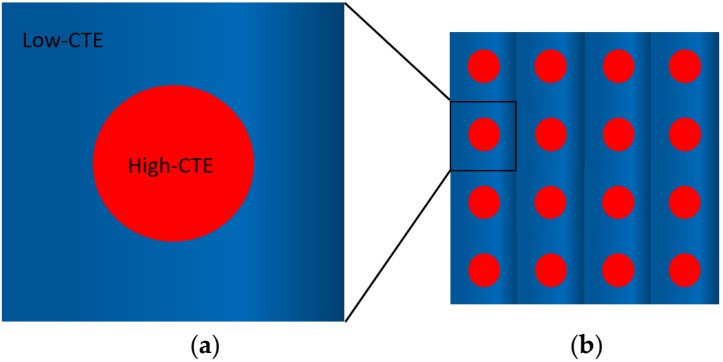

(**a**)　　　　　　　　　　　　　　　　　　(**b**)

**Figure 2.** A concept of a single-unit cell (**a**) and a 4 × 4 cell lattice structure (**b**).

The design of the lattice outer part is illustrated in Figure 3. The design comprised four CTE-minimising layers, each connected to a load-bearing pillar along one of its edges. The deformation mechanism section was designed as a hollow octagonal prism with eight connecting struts, intended to bend when the inner part expanded in response to increased temperature. This bending provides a complementary deformation that minimises the structure's planar CTE. The length of the hollow octagonal prism (in the *Y*-axis) was 0.94 *L*, where *L* is the lattice side length. The prism is offset (in the *Y*-axis) by 0.03 *L* from both the top and bottom surfaces. In addition, the lattice was designed to expand along the diagonal directions, as shown in Figure 3b, providing more space for the load-bearing pillars at the cell's corners where they can be thicker and stiffer than they would be if they were placed at the cell's faces.

The four CTE-minimising layers had different orientations around the *Y*-axis: 0°, 90°, 180°, and 270°, as illustrated in Figure 3b. The layers formed two pairs, according

to their connectivity with the load-bearing pillars along the cell's edges. The two layers connected to diagonally-opposite pillars were connected by the struts of the octagonal prism discussed above.

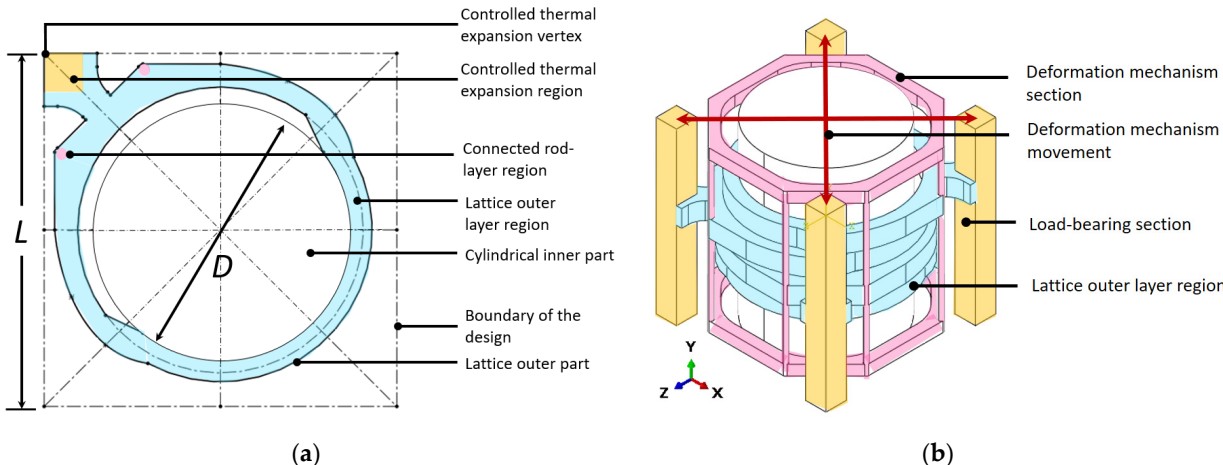

(a)      (b)

**Figure 3.** The concept of low-CTE design. (**a**) Top view of a drafted layer, where *L* is the original length of the lattice and D is the original diameter of the cylindrical inner part; (**b**) Isometric view of a unit cell (the two-way arrows indicate the deformation movement of the lattice).

Tessellation of the unit cell in order to construct a lattice structure was then investigated. This posed a challenge because the unit cell was not rotationally symmetric in the XZ plane, resulting in different CTEs in each direction; however, it is desirable to engineer a structure with isotropic planar CTE. In Figure 4, the layers in the unit cell are numbered according to the position of the load-bearing pillar (top view of Figure 4a). Layers 1, 2, 3, and 4 (front view of Figure 4a) are the lowest layer, the second-lowest layer, etc., respectively. The symbol 'P' was used to represent the orientation of the unit cell, as shown in Figure 4b. Five potential patterns were selected and created by changing the orientation around the *Y*-axis and combining them into $2 \times 1 \times 2$ lattices, as shown in Figure 4b. We investigated the 1st pattern to demonstrate the CTEs of a tessellated lattice and the difference between the mean CTEs in the X- and Z-axes of each composed lattice. A $1 \times 1 \times 1$ lattice up to a $4 \times 1 \times 4$ lattice (Figure 5) were chosen for this investigation.

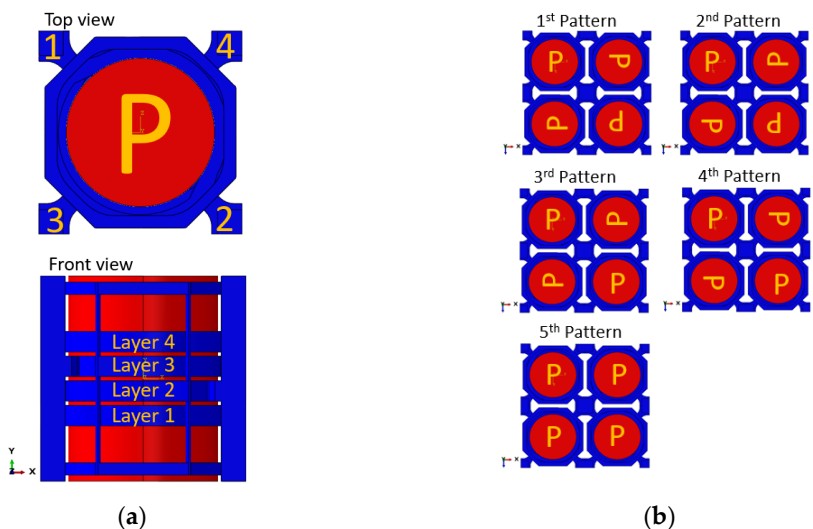

(a)      (b)

**Figure 4.** Annotation and patterns of the lattices. (**a**) Top view and front view of the unit cell; (**b**) Patterns of $2 \times 1 \times 2$ lattices.

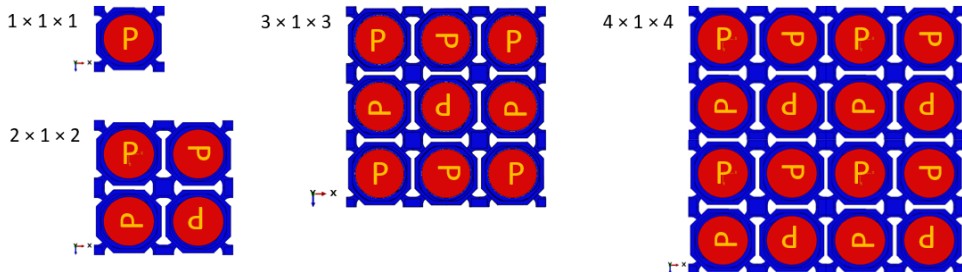

**Figure 5.** Illustration of $1 \times 1 \times 1$, $2 \times 1 \times 2$, $3 \times 1 \times 3$ and $4 \times 1 \times 4$ lattices.

### 2.2. The Finite Element Method

Concept lattice designs were modelled in SolidWorks 2018 to obtain representative CAD models, and their performances were simulated using the FE method implemented in Abaqus 2018 and 2020. The initial lattice unit cell dimensions (i.e., in their un-heated state) were $(20 \times 20 \times 20)$ mm. The temperature of all elements in all models was raised from 19–23 °C, which represents the temperature variation in a typical temperature-controlled metrology laboratory. All nodes on three of the lattice's cubic edges were constrained in space, as shown in Figure 6, to enable the calculation of thermally induced expansion. The XSYMM (Symmetry about a plane X = constant (U1 = UR2 = UR3 = 0)), YSYMM (Symmetry about a plane Y = constant (U2 = UR1 = UR3 = 0)), and ZSYMM (Symmetry about a plane Z = constant (U3 = UR1 = UR2 = 0)) boundary conditions were applied to the nodes on the surfaces of the YZ plane in the –X direction (labelled 'YZ $(-X)$' in Figure 6), the XZ plane in the –Y direction ('XZ $(-Y)$'), and the XY plane in the –Z direction ('XY $(-Z)$'). The thermal expansion of the lattice was calculated by examining the displacement of the nodes on the surfaces opposite to those that were constrained. The in-plane CTE ($\alpha_{X,Z}$) was found by taking the average of the CTEs in *X*-axis and *Z*-axis over four orientations of the unit cell or lattice around the *Y*-axis: 0° (origin), 90°, 180°, and 270°. In a preliminary investigation, the CTE of a unit cell was calculated using the mesh dependency at different mesh densities to determine the required mesh for accurate CTE modelling. The elements used were ten-node tetrahedral elements (C3D10) because, in comparison with hexagonal elements, they provided a more accurate description of the complex lattice geometry [34,35]. The number of elements per unit cell was increased from 200,000 to 1,100,000 as shown in Figure 7. A mesh density of 440,000 elements per unit cell was sufficient to determine the CTE of the lattice with a deviation of less than −0.4% from the 1,100,000 element result, indicating the result is well converged. Figure 6 shows the unit cell model meshed with 440,000 elements.

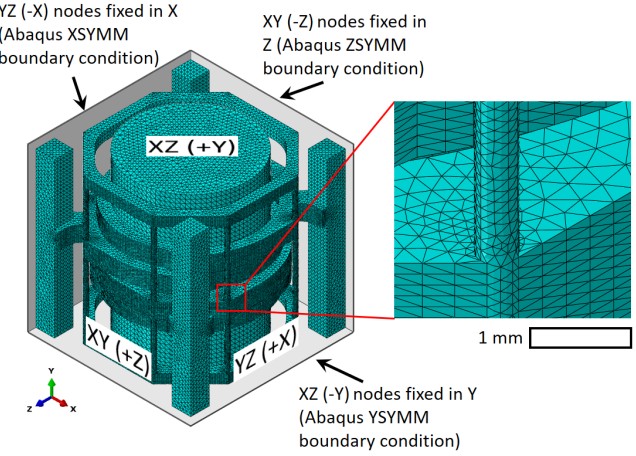

**Figure 6.** Representative example of the converged mesh of the low-CTE concept lattice unit cell and its boundary conditions applied on a lattice unit cell design.

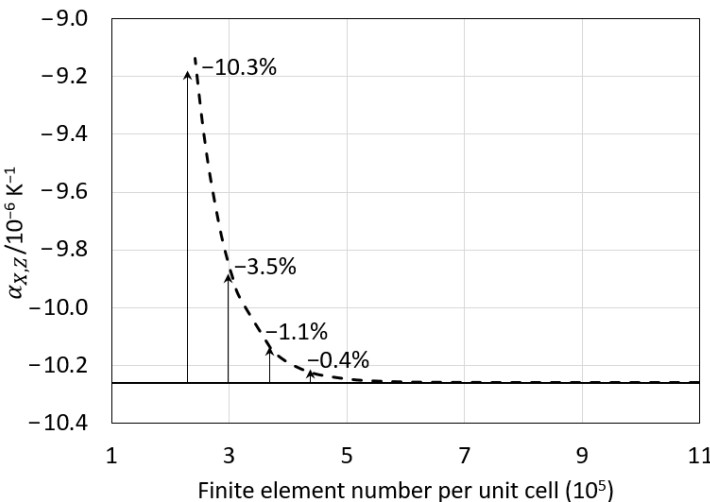

**Figure 7.** Finite element mesh density convergence for the low-CTE lattice cell.

## 3. Results

### 3.1. Low Planar CTE Lattice Results

The concept design presented in the previous section has several design parameters that may be varied to control the resulting structure's CTE. These include the original length of the lattice $L$, the original diameter of the cylindrical inner part $D$, and the ratio between the CTEs of the cylindrical inner part material and the lattice outer part material $\alpha_2/\alpha_1$. The ratios $D/L$ and $\alpha_2/\alpha_1$ were varied, respectively, from 0.30 to 0.73 (the maximum ratio that the lattice was designed) and from 1.10 to 2.20. Examples of the resulting lattice cells are shown in Figure 8.

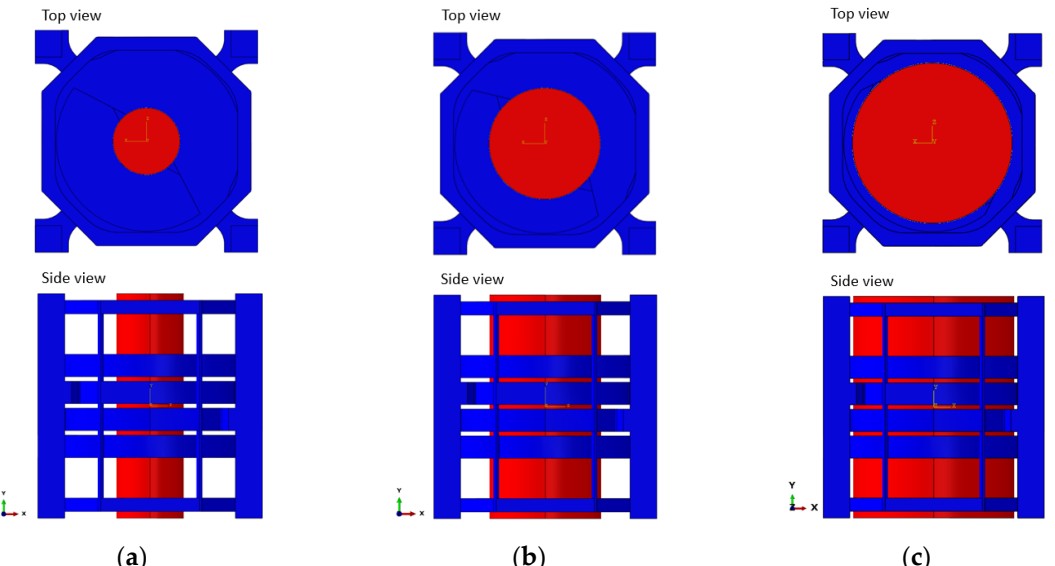

**Figure 8.** Illustration of the top view and the side view of (**a**) the $(D/L) = 0.30$ model; (**b**) the $(D/L) = 0.50$ model; and (**c**) the $(D/L) = 0.73$ model.

Figure 9 shows that the concept lattice unit cell provides in-plane CTEs ($\alpha_{X,Z}$) from approximately $-62 \times 10^{-6}$ K$^{-1}$ to $96 \times 10^{-6}$ K$^{-1}$ by varying the design parameters. $\alpha_{X,Z}$ data were fitted with a first order polynomial surface function of the form

$$\alpha_{X,Z} = a + b\left(\frac{\alpha_2}{\alpha_1}\right) + c\left(\frac{D}{L}\right) \tag{1}$$

which provides a guide allowing designers to customise the CTE of the proposed lattice structure. The fit provides an accurate description of the data (with $R^2 = 0.994$) over the examined range of $D/L$ and $\alpha_2/\alpha_1$.

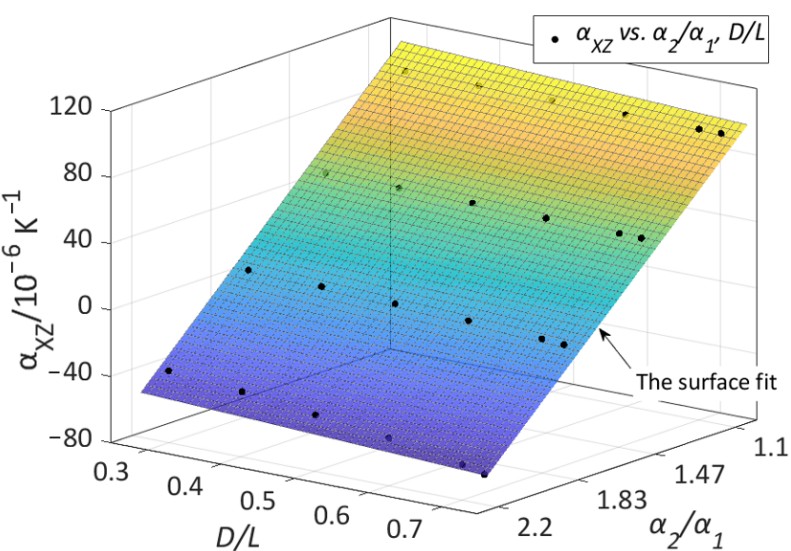

**Figure 9.** The low planar CTE lattice results and its surface fit, where $L$ is the original length of the lattice and $D$ is the original diameter of the cylindrical inner part, and $\alpha_2/\alpha_1$ is the ratio of CTEs of the cylindrical inner part material and the lattice outer part material.

The resulting fit parameters, with 95% confidence bounds, were found to be $a = (253 \pm 10) \times 10^{-6}$ K$^{-1}$, $b = (-132 \pm 5) \times 10^{-6}$ K$^{-1}$, and $c = (-20 \pm 10) \times 10^{-6}$ K$^{-1}$, which can henceforth be used to determine the $D/L$ and $\alpha_2/\alpha_1$ parameters to achieve a pre-defined CTE.

The CTEs of Nylon 12 and UHMWPE are $109 \times 10^{-6}$ K$^{-1}$ and $200 \times 10^{-6}$ K$^{-1}$ [36], respectively. The ratio of $\alpha_2/\alpha_1$ for our chosen materials was, therefore, approximately 1.83, meaning that a unit cell made of such materials can provide CTEs between $7.8 \times 10^{-6}$ K$^{-1}$ and $1 \times 10^{-9}$ K$^{-1}$.

### 3.2. Pattern Selection for Low-CTE Lattice Design

When a $(20 \times 20 \times 20)$ mm single-unit cell was examined using the FE method described in Section 2.2, there were differences between CTEs in the X- and Z-axes. This anisotropic thermal expansion would be undesirable for the end-application of the precision machine frame, where predictable response is paramount. One solution to decrease the differences between CTEs in the X- and Z-axes was to compose the unit cells in an arrangement that compensated for the anisotropy. The in-plane CTE (in the XZ plane) was investigated for five unique orientation patterns (see Figure 4). The 1st pattern lattice provided the lowest in-plane CTE ($5.4 \times 10^{-8}$ K$^{-1}$), while the 4th pattern gave the highest in-plane CTE ($29.0 \times 10^{-8}$ K$^{-1}$), as shown in Table 2.

**Table 2.** The in-plane coefficient of thermal expansion of designs of the $2 \times 1 \times 2$ lattice.

| Pattern | In-Plane CTE ($10^{-8}$ K$^{-1}$) |
|---|---|
| 1 | 5.4 |
| 2 | 12.8 |
| 3 | 8.9 |
| 4 | 29.0 |
| 5 | 9.5 |

Table 2 shows the in-plane CTE results of five unique orientation patterns. The 1st pattern was the lattice that was composed of the 0°, 90°, 180°, and 270° counterclockwise orientated unit cells in the top left, top right, bottom right, and bottom left cells, respectively. The 1st pattern provided the lowest in-plane CTE because it was the only pattern with connecting layers in a single plane (the XZ plane), see Figure 10c, so it means that the displacement would occur only in the XZ plane. The deformation of the lattice due to the expansion of the inner cylindrical part was, therefore, constrained to a single plane. In Patterns 2 to 5, they provided higher in-plane CTEs because the arrangement of the connecting layers were out of the XZ plane, resulting in displacements coming out of the XZ plane (see Figure 10d). Moreover, when the 1st pattern lattice was tessellated over a greater numbers of cells, it was observed that even numbers of unit cells provided low CTE anisotropy ($3 \times 10^{-8}$ K$^{-1}$ for the $2 \times 1 \times 2$ lattice and $1 \times 10^{-8}$ K$^{-1}$ for the $4 \times 1 \times 4$ lattice), as shown in Table 3, because they consisted of $2 \times 1 \times 2$ lattices which constrained the deformation to a single plane, e.g., $4 \times 1 \times 4$ lattice was composed of four sets of $2 \times 1 \times 2$ lattice. However, this paper focuses on design concepts, and results from larger lattices with more cells require considerably greater computational resources than could be employed here.

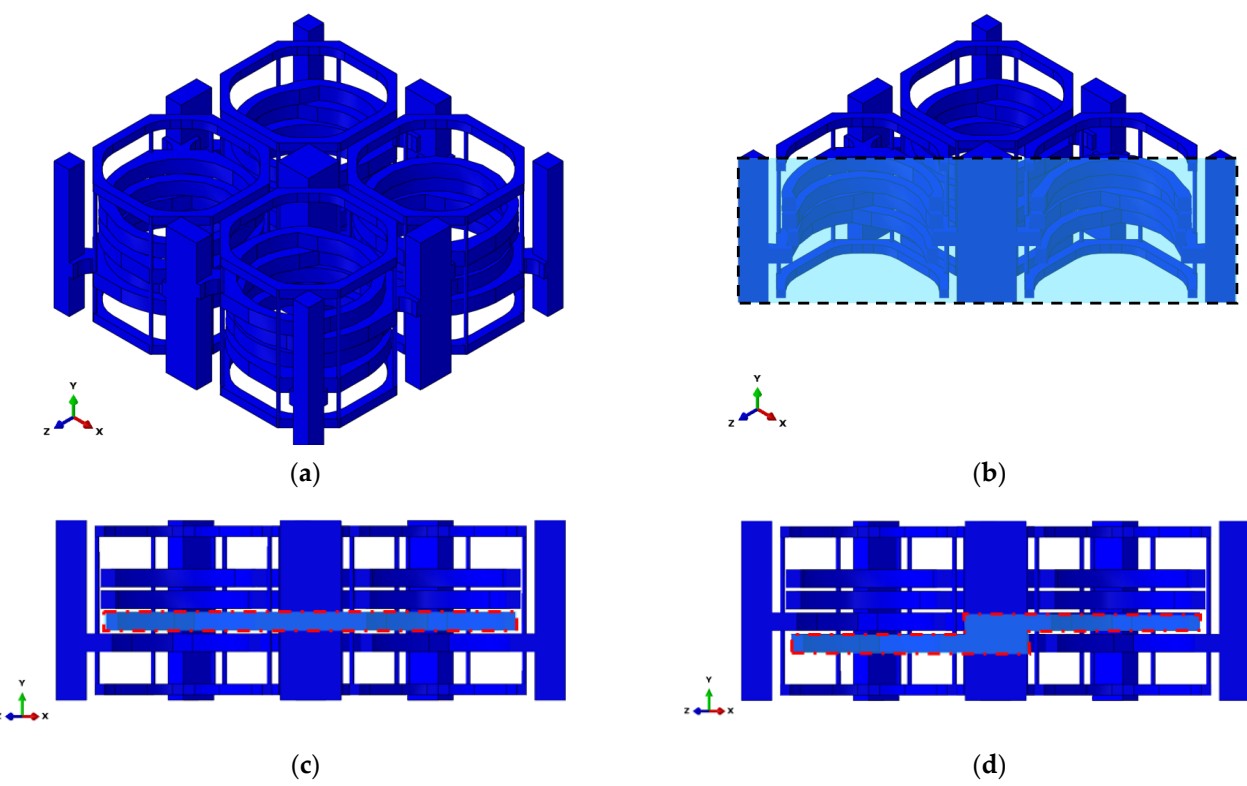

**(a)**      **(b)**

**(c)**      **(d)**

**Figure 10.** The effect of layer alignment on the deformation of the $2 \times 1 \times 2$ lattice outer part. (**a**) shows an example of the 1st pattern; (**b**) shows an example of the cross-section of the 1st pattern; (**c**) shows an example of the cross-section of the 1st pattern at Layer 2; and (**d**) shows any of the other patterns' connecting layers.

**Table 3.** The in-plane coefficient of thermal expansion of designs.

| Lattice | In-Plane CTE ($10^{-8}$ K$^{-1}$) | In-Plane CTE Anisotropy ($10^{-8}$ K$^{-1}$) |
| --- | --- | --- |
| $1 \times 1 \times 1$ | 0.1 | 1379 |
| $2 \times 1 \times 2$ | 5.4 | 3 |
| $3 \times 1 \times 3$ | 8.7 | 149 |
| $4 \times 1 \times 4$ | 0.03 | 1 |

## 4. Discussion

The CTE results for the design in this paper showed that the planar CTE of the proposed lattice design can be tailored by changing the ratios of $D/L$ and $\alpha_2/\alpha_1$. The selection of $D$ and $L$ was carried out by modifying the geometry with CAD software. The parameters $\alpha_1$ and $\alpha_2$ were varied by the selection of the lattice outer part and the cylindrical inner part materials. However, $\alpha_2$ should be greater than $\alpha_1$ to make the deformation mechanism properly work.

Based on the relationship of the cylindrical inner part and the lattice outer part using parametric constraints, when the size of the cylindrical inner part is increased, the internal geometry of the outer part must increase to maintain physical contact. This leads to varying quantities of material in each part, as shown in Figure 11. In turn, this will affect the manufacturing cost of the lattice structure, since AM processes are relatively costly, especially LPBF [7]. It is clear that the cost can be reduced by designing the lattice to have a ratio of $D/L$ as large as possible in this structure; however, for a target CTE of zero, this places restrictions on the CTEs of the selecting materials.

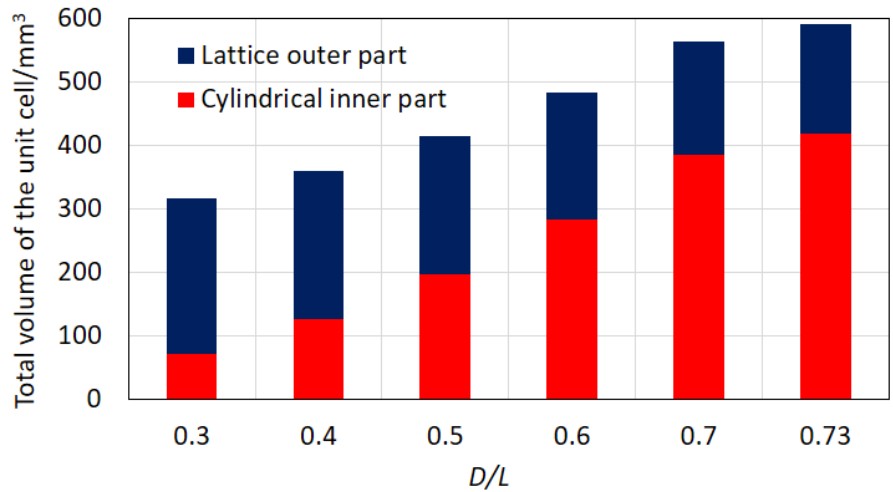

**Figure 11.** The total volume of the $(10 \times 10 \times 10)$ mm unit cell by varying the ratio of $D/L$ between 0.3 and 0.73.

Using a similar assembly technique to that examined here, Jefferson et al.'s structure [16] provided near-zero CTE, while the design in this paper can provide a wider range of CTE including near-zero, achieved with $D/L = 0.73$ and $\alpha_2/\alpha_1 = 1.83$. Although other polymer-based structures could provide wider ranges of CTEs than the lattice in this paper (i.e., $-434 \times 10^{-6}$ K$^{-1}$ to $396 \times 10^{-6}$ K$^{-1}$ from Akihiro et al. [12] and $-300 \times 10^{-6}$ K$^{-1}$ to $1000 \times 10^{-6}$ K$^{-1}$ from Talezawa and Kobashi [13]), the internal geometries of those structures were more complex, required additional topology optimisation software to model the structures, and still needed an advanced multi-material AM process to fabricate. The structure with the chosen parameters could provide a CTE of $1 \times 10^{-9}$ K$^{-1}$, which was a much lower CTE than many commercial instruments, such as $23 \times 10^{-6}$ K$^{-1}$ for aluminium 7075's CTE and $10 \times 10^{-6}$ K$^{-1}$ for stainless steel 431's CTE [25] of the working surface of commercial optical breadboards.

Moreover, the 1st pattern cell tessellations composed of even numbers in both the X- and Z-axis are recommended for low-CTE applications; however, further investigations are required to identify and ideally exploit the cause. An added advantage of charactering CTE via the fitted surface shown in Figure 9 is that adjustments can be made for variations in material properties or geometrical accuracy with a small number of calibration experiments.

## 5. Conclusions

The concept introduced in this paper aimed to create a planar near-zero thermal expansion lattice structure and the designs showed that the thermal expansion could be controlled, by the deformation mechanism in internal geometry, using a combination of two positive homogenous materials. The design introduced the separation of load-bearing sections and deformation mechanism sections and showed the optimised functions in the sections of internal geometry to increase their performances. It was shown that the CTE could theoretically be reduced to $1 \times 10^{-9}$ K$^{-1}$, which was the nearest to the near-zero CTE that could be achieved from the design.

In summary, the designs illustrated how lattices could be created to obtain a wide range of CTEs by changing the ratios of the diameter of the cylindrical inner part over the length of the lattice, and the ratio between the CTEs of the cylindrical inner part material and the lattice outer part material. Then, pattern selection was introduced to reduce differences between CTEs in the X- and Z-axes of the lattices that compensated for the anisotropy of the lattice.

## 6. Future Work

In future, the design will be fabricated using metal to prevent distortions of the frame due to the weight of workpieces and precision measurement equipment [6]. Moreover, the lattice needs to be tested in a physical experiment before being implemented in a real application. Figure 12 shows examples of the optimised design specimens in $(40 \times 40 \times 40)$ mm and $(50 \times 50 \times 50)$ mm. The lattice outer parts composed of Nylon 12 were fabricated using an EOS Formiga P100 PBF machine. Cylindrical inner parts composed of UHMWPE were manufactured and resized by conventional manufacturing processes.

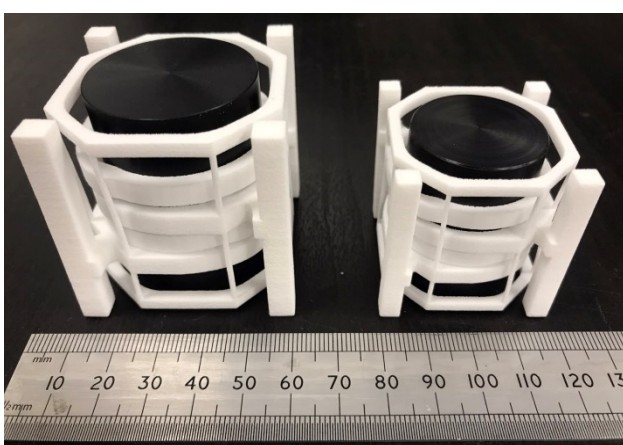

**Figure 12.** Examples of the design specimens in several sizes.

**Author Contributions:** Conceptualization, P.J.; methodology, P.J.; software, P.J.; validation, P.J.; formal analysis, P.J. and I.M.; investigation, P.J. and I.M.; writing—original draft preparation, P.J.; writing—review and editing, P.J., I.M., I.A. and R.L.; visualization, P.J.; supervision, I.M., I.A. and R.L. All authors have read and agreed to the published version of the manuscript.

**Funding:** This research received no external funding.

**Institutional Review Board Statement:** Not applicable.

**Informed Consent Statement:** Not applicable.

**Data Availability Statement:** The data presented in this study are available on request from the corresponding author.

**Conflicts of Interest:** The authors declare no conflict of interest.

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
