# Peer review of "Low Thermal Expansion Machine Frame Designs Using Lattice Structures"

_applsci, doi:10.3390/app11199135_

Round 1
Reviewer 1 Report
The manuscript entitled “Low thermal expansion machine frame designs using lattice” dealing with directed energy deposition structures” was reviewed. The paper adds interesting material to the body of DFAM and is very useful.
Please see my comments below in improving the quality of the paper:
- Authors are encouraged to proofread the paper
- Page 7 line 181 the reference is not clear. Seems the endnote has the issue. Please correct this.
- Add a sentence about the dependency of the simulation to the mesh.
- Authors should add a section at the end of the paper about the future work.
- Authors should read and add the following new and novel references from 2021:
C Pan, Y Han, J Lu, Design and optimization of lattice structures: A review
Qin, Q., J. Huang, J. Yao, and W. Gao, Design and optimization of projection stereolithography additive manufacturing system with multi-pass scanning.
Riedle, H., A. Ghazy, A. Seufert, V. Seitz, B. Dorweiler, and J. Franke, Generic design of an anatomical heart model optimized for additive manufacturing with silicone.
Lopez Taborda, L.L., H. Maury, and J. Pacheco, Design for additive manufacturing: a comprehensive review of the tendencies and limitations of methodologies.
Gonzalez Alvarez, A., P.L. Evans, L. Dovgalski, and I. Goldsmith, Design, additive manufacture and clinical application of a patient-specific titanium implant to anatomically reconstruct a large chest wall defect.
Reviewer 2 Report
The paper can be improved. The method may be interesting but not be present well.
Reviewer 3 Report
Dear authors, your manuscript on low thermal expansions machine frame designs using lattice structure was a very well presented and written.
With your finite element investigation, knowledge of the coefficients of thermal expansion and using additive manufacturing, you were able to optimize the design of a low-CTE structure, that measures tens of centimeters in height. This design is promising and should be pursued further.
Your introduction was very well written, with good references and explaining the difficulty in achieving low-CTE materials for engineering applications such as welding. You also explained what was done in the past and what your new approach changes in advanced manufacturing.
Here are a few questions:
- lines 120-122: Is this done by hand or by machine? Please specify.
- There are missing references to figures throughout the manuscript
- line 170: please explain the XSYMM, YSYMM & ZSYMM acronyms.
- line 197-198: how were those values chosen?
- lines 225-226: Why do these variations occur?
- lines 235-236: can you explain how this constrain affects the calculations and why it is "good"?
- line 237: "greater numbers" how many did you test?
- line 238: I have not seen an explanation on why even numbers of cells provide lower CTE anisotropy
- line 247: What are the limit conditions of those ratios?
- line 257: what are the restrictions and why?
- Is figure 12 the actual piece with the correct material or was it 3D printed for visual information only?
I believe addressing these questions simply in the text will enhance an already good manuscript.
Congratulations for your research.
Reviewer 4 Report
Page 5, Line 149 and line 150, (top view of Error! Reference source 149 not found.a) needs to be fixed.
Page 6, Line 152 and line 154, (Error! Reference source not found.b.) needs to be fixed.
